# Development and piloting of a survey to estimate the frequency and nature of potentially harmful preventable problems in primary care from a UK patient's perspective

Susan J Stocks,[1] Ailsa Donnelly,[2] Aneez Esmail,[1] Joanne Beresford,[2] Carolyn Gamble,[2] Sarah Luty,[3] Richard Deacon,[4] Avril Danczak,[5] Nicola Mann,[2] David Townsend,[2] James Ashley,[6] Paul Bowie,[7,8] Stephen M Campbell[1]

For numbered affiliations see end of article.

**Correspondence to**
Dr Susan J Stocks;
sjstocks@btinternet.com

## ABSTRACT

**Objectives** To design and pilot a survey to be used at the population level to estimate the frequency of patient-perceived potentially harmful preventable problems occurring in UK primary care. To explore the nature of the problems, patient-suggested strategies for prevention and opinions of clinicians and the public regarding the potential for harm.

**Design** A survey was codesigned by three members of the public and one researcher and piloted through public and patient involvement and engagement networks.

**Setting** Self-selected sample of the UK population.

**Participants** 977 members of the public accessed the online survey during October and November 2015.

**Primary outcome measures** Respondent feedback about the ease of completion of the survey, quality of responses in terms of review by clinicians and members of the public, preliminary estimates of the frequency and nature of patient-perceived potentially harmful problems occurring in the last 12 months.

**Results** 638 (65%) members of the public completed the survey and few respondents reported any difficulty in understanding or completing the survey. 132 (21%) respondents reported experiencing a potentially harmful preventable problem during the past 12 months and 108 (82%) of these respondents provided a description that was adequate for at least one clinician to form an opinion about the potentially harmful problem. Respondents were older than the UK generally, more likely to work or volunteer in the healthcare sector and tended to use primary care more frequently but their confidence and trust in their own general practitioner (GP) was similar to that of the UK population as measured by the annual English GP patient survey.

**Conclusions** The survey was acceptable to patients and mostly provided data of sufficient quality for review by clinicians and members of the public. It is now ready to use at a population level to estimate the frequency and nature of potentially harmful preventable problems in primary care from a patient's perspective.

## Strengths and limitations of this study

► We have designed and tested a survey to measure the frequency and nature of potentially harmful preventable problems in primary care from the patient's perspective.
► The survey was codesigned by three members of the public and piloted through extensive public and patient involvement (PPI).
► The patient-described scenarios were reviewed by primary care clinicians.
► The study respondents were self-selected through PPI and engagement groups.
► The survey is ready to be administered to a representative sample of the general population.

They tend to view safety in terms of the overall balance of benefit and harm over time whereas healthcare professionals often see high-quality healthcare occasionally punctuated by safety incidents and adverse events.[2] Furthermore, patients may hold different opinions about how to improve patient safety[3 4] or different priorities to clinicians, for example, identifying psychological and emotional harm rather than technical errors.[5] Involving patients in identifying errors and reducing harm occurs in secondary care,[6] but patient-reported outcomes can show poor concordance between patients and clinicians, for example, in reporting adverse symptom events in the context of drug safety.[7] Nonetheless, patients are thought to be capable of reporting medical errors accurately.[6 8] Involving patients is advocated as a way to improve safety,[9] and this approach would be facilitated through patients and professionals having an understanding of each other's expectations and priorities.

## BACKGROUND

Patients are thought to take a different view of patient safety to healthcare professionals.[1]

Studies that quantify patient safety problems in primary care are uncommon and incidence estimates from record review or incident reporting by clinicians range from less than 1 to 24 per 100 consultations or record review.[10–12] The National Reporting and Learning System in England and Wales records patient safety incidents reported by healthcare professionals; only 1% of these reports originate from primary care[13] which likely reflects under-reporting.[14 15] Still fewer studies have quantified patient safety problems in primary care from the patient's perspective.[16] A 2013 European survey of the UK public reported that 43% of respondents felt that it was 'likely' that patients could be harmed by non-hospital healthcare, an increase from 37% in 2009.[17] In Norway, a population-level survey found that the patient-reported lifetime probability of ever experiencing an adverse event was 10%, of which around two-thirds of respondents attributed the cause of their event as their general practitioner (GP).[4] In Spain, a telephone survey of patients estimated that around 7% of patients experienced a self-reported adverse event during a 1-year period.[18] A USA practice-based website observed an incidence rate of patient-reported adverse events of 1.4% over 2 years.[19] Data from the UK are sparse; this may be partly due to the lack of a valid and reliable instrument to make a comprehensive measurement of safety in primary care.[20] The Patient Reported Experiences and Outcomes of Safety in Primary Care questionnaire should help to address this knowledge gap.[21 22]

Although it is acknowledged that patients tend to take a different view to professionals,[1 2] most research into patient safety is initiated by clinicians with patients invited to contribute. We choose to take an alternative approach whereby the study design was conceived, designed and implemented by a team of three members of the public and one researcher with primary care professionals being invited to contribute later. Previous work has shown that patient-initiated surveys can provide meaningful feedback and guide improvements.[23] Our aim was to design a survey asking about potentially harmful preventable problems occurring in UK primary care in partnership with the Greater Manchester Primary Care Patient Safety Translational Research Centre Research User Group, (GMPSTRC RUG), a public and patient involvement (PPI) group.[24] Specifically, we aimed to:

1. codesign (with PPI partners) and test a survey asking about problems occurring in primary care that caused, or had the potential to cause, preventable harm as perceived by patients;
2. pilot the survey to examine the usefulness and overall quality of the information collected with respect to describing the patient-perceived problems, the primary care service involved, how the problem was discussed (if it was) and how it might have been prevented;
3. compare the opinions of the survey respondents, members of the public and primary care clinicians as to the likelihood the patient-reported scenario describes a potentially harmful preventable problem.

> **Box    Brief summary of questionnaire (see online supplementary appendix 1 box A for full version of survey)**
>
> 1. Did you have confidence and trust in the GP you saw or spoke to at your last appointment? (benchmarking question)
> 2. When using primary care, have you ever felt concerned that your health might be worsened, or actually was made worse, because of a mistake or a problem that could have been prevented? If no to Q10, if yes to Q3
> 3. How long ago did the mistake or preventable problem happen?
> 4. How did this affect your health?
> 5. Which primary care service were you using when the mistake or preventable problem occurred?
> 6. Briefly describe the mistake or problem and how it happened.
> 7. Could the mistake or problem have been avoided? If so, how?
> 8. Were you able to talk about the mistake or problem with anybody working in the primary care service? If not, why not?
> 9. If you discussed the mistake or problem with somebody working in primary care, please describe their job or role.
> 10. In the list below are some examples of preventable problems* that might happen when using primary care. Has anything similar happened to you in the last 12 months? If yes, go to Q4.
>
> *See Q10 in online supplementary appendix 1 box A for the list of preventable problems.

## METHODS
### Designing and piloting of the survey (aim 1)
Our main aim was to design a survey asking about problems occurring in primary care that caused, or had the potential to cause, preventable harm as perceived by patients that was easily understood and free from jargon. Currently, there is no well-established terminology for asking such a question.[8] The process began with a discussion between three members of the GMPSTRC RUG (AD, JB, CG) and one academic researcher (SJS). Questions used in previous surveys addressing a similar question[4 17–19] were shared among the project team and used to generate several candidate questions. These questions were then discussed privately among the project team's friends and family and within the project team (SJS, AD, JB, CG). The discussion was facilitated by making the candidate questions available online. After two iterations of this process. the survey (see box and online supplementary appendix 1 box A) was piloted online through newsletters or group mailings of several PPI and public engagement networks during November and December 2015. These networks were the associate GMPSTRC RUG, the Public Programmes team at Central Manchester Foundation Trust, the Citizen Scientist project, the Primary Care Research in Manchester Engagement Resource, North West People in Research Forum and Help Beat Diabetes volunteers (details of these groups and networks are provided in online supplementary appendix 1 box B).

The first question (Q1, box) was taken from the English GP patient survey in order to compare the overall level of confidence and trust in their GP among the survey respondents with that across England.[25] The second question (Q2 in box) is the main screening question; those responding negatively

to Q2 (ie, not experienced a preventable-problem) were directed to a more specific question with a list of commonly understood patient safety events (Q10, online supplementary appendix 1 box A). If this prompted recognition of experiencing a potentially harmful preventable problem, they were returned to Q4 (box). The rationale behind this approach was that the screening question (Q2, box) should be non-leading and encourage the respondents to describe their preventable problems through the subsequent questions without the suggestion that inevitably occurs following a list of possible potentially harmful preventable problems. However, if the respondent did not believe that they had experienced a potentially harmful preventable problem, then the prompt question (Q10, box) would ensure that this was the case and also test the sensitivity of Q2 (box). The option to answer on behalf of a friend or relative was offered to those who have not had a personal experience to report. This was to ensure sufficient responses to adequately test the questionnaire and also to discourage respondents from answering with another person's experience as their own. Respondents were also asked whether they worked or volunteered in the healthcare profession and to comment on the ease of completion of the questionnaire.

### Coding of reported events (aims 2 and 3)
#### Type of problem (aim 2)
The nature of the problem in each described scenario was coded at face value, that is, as the patient described without further interpretation, by one author (SJS) and checked by a second author (JA for dental scenarios, PB for all other scenarios). A bottom-up (inductive) approach was used to identify similar topics which were coded then cross-matched to an existing taxonomy for errors in general practice[26 27] (online supplementary appendix 1 table A). All the new codes matched the existing taxonomy within the higher two levels and the medication-related scenarios were coded to a finer level (online supplementary appendix 1 table B).

#### Likelihood the scenario described a potentially harmful preventable problem (aim 3)
Five GPs, one general dental practitioner and seven members of the public estimated the likelihood that, in their opinion, each patient-described scenario was a potentially harmful preventable problem. Brief biographies of the coders are provided in online supplementary appendix 1 table C. Some examples of the information provided to the coders are shown in boxes 1–23 in online supplementary appendix file 2 and consisted of the responses to Q5 to Q9 (box). They were not given any demographic information or the patient's estimate of the impact on their health (Q4, box). Coders were asked to score each scenario from very likely (5) to definitely not (1) in response to the question 'How likely do you think it is the patient was correct in thinking that their health might be worsened, or actually was made worse, because of a mistake or a problem in primary care that could have been prevented?' Coders could also respond 'insufficient information', 'Don't know' and give free text

feedback (online supplementary appendix 1 table D). The clinician scores were used to categorise the scenarios into groups with higher or lower estimated likelihoods that they were a potentially harmful preventable problem as below.

► Higher threshold: Median score of 5 ('very likely or certain') or 4 ('probably') or at least one score of 5 ('very likely or certain').
► Lower threshold: Median score of 3 ('possibly') or at least one score of 4 ('probably' or higher).
► All other scenarios: Median score below 3 ('possibly') and zero scores above 3 ('possibly').

### Statistical analysis
Simple cross-tabulations were used to describe the data and a binary logistic regression model was used to explore whether particular types of patient were more likely to perceive potentially harmful preventable problems, for example, by demographics or their opinions. Comparisons between demographics and outcomes for the respondents and the UK (or England) population were made using a $\chi^2$ test. All analyses were done using Stata V.14.

### Public and patient involvement
PPI was central to this codesign study and was provided through the GMPSTRC RUG[24] and other PPI networks (online supplementary appendix 1 box B). The study was conceived, designed, implemented and analysed by a team of three members of the public (AD, CG, JB) and one researcher (SJS). At the outset, the researcher presented the existing literature on this topic to the PPI members of the research team who then codesigned the first draft of the survey which was tested through the PPI members' personal contacts. The piloting of the survey was through existing PPI networks as listed in online supplementary appendix 1 box B. The scoring of the questions as to the likelihood they described a potentially harmful preventable problem was undertaken by seven members of the public, two of whom had no previous experience in PPI (as well as five GPs and one general dental practitioner as described in online supplementary appendix 1 table C). These findings will be disseminated to all the PPI groups that contributed to the pilot study and the authors will forward these results to their personal contacts who contributed to the questionnaire design.

### RESULTS
#### The survey design (aim 1)
The involvement of the PPI partners in the survey design had a profound impact on the piloted version of the survey. Professional researchers may have focused more on asking questions in a way that forces the responses into categories but the PPI partners were more concerned that respondents should have the freedom to express themselves and the categorisation should occur during the analysis. They themselves had often completed surveys where there was no appropriate option in the categorical responses. We did not find any of the previous approaches[4 17–19] suitable for this survey and chose to design a new question. The best option was felt to

be an open question with a prompt question for individuals who did not recognise the concept of a preventable potentially harmful event. Another point of debate was whether we should ask initially about a 'problem' then ask if it was 'preventable' in a second question. The difficulty with simply asking about a 'problem' is that most patients visit their GP because they have a health problem; therefore, we thought it was more practical to focus immediately on the concept of a preventable problem encapsulated in a single phrase with a back-up question to ensure it was indeed preventable.

### Ease of use of the survey (aim 1)

Over 250 respondents provided free text feedback on the survey, 200 comments reported that the questionnaire was easy to complete and understand and just one comment described the survey as complex. Most of the remaining comments expressed the desire to be able to provide more

information, for example, more than one event or report for a relative or as a carer (reporting on behalf of another person was excluded for events occurring more than 12 months ago) and 13 comments actually provided this unrequested information. Nobody used the 'Do not understand the question' option as their response to Q2 of box. A few respondents found it difficult to find a suitable option to describe their pattern of use of primary care or their role as a worker or volunteer in healthcare. Demographic information was not provided by 83 (13%) respondents, possibly due to lack of clarity about the end of the survey since they completed all other questions.

### Summary statistics (aim 2)

In total, 977 members of the public accessed the online pilot survey and 638 (65%) completed the survey during October and November 2015. The majority of

**Table 1** Characteristics of survey respondents

| Variable | All respondents n=638 | Ever had problem n=223 | Had problem in last 12 months n=132 | UK population comparator |
|---|---|---|---|---|
| GP satisfaction | Missing=0 | Missing=0 | Missing=0 | English GP patient survey[25] |
| Yes, definitely | 384 (60%) | 81 (36%) | 55 (42%) | 64% |
| Yes, to some extent | 208 (33%) | 110 (49%) | 52 (39%) | 28% |
| No, not at all | 39 (6%) | 27 (12%) | 21 (16%) | 4% |
| Do not know/cannot say | 7 (1%) | 5 (2%) | 4 (3%) | 3% |
| Worked or volunteered in healthcare | Missing=92 | Missing=40 | Missing=19 | NHS workforce* |
| Yes | 166 (30%) | 64 (35%) | 41 (36%) | 3% |
| Gender | Missing=87 | Missing=38 | Missing=16 | ONS mid-2015 estimates† |
| Female | 268 (49%) | 106 (57%) | 63 (54%) | 51% |
| Age | Missing=85 | Missing=37 | Missing=15 | ONS mid-2015 estimates† |
| 16–34 years | 42 (8%) | 22 (12%) | 11 (9%) | 31% |
| 35–54 years | 143 (26%) | 54 (29%) | 34 (29%) | 34% |
| 55–64 years | 162 (29%) | 59 (32%) | 31 (27%) | 14% |
| 65–74 years | 170 (31%) | 44 (24%) | 32 (27%) | 12% |
| Over 75 years | 36 (7%) | 7 (4%) | 9 (8%) | 9% |
| Last primary care contact | Missing=88 | Missing=39 | Missing=14 | English GP patient survey[25] |
| Within last week | 169 (31%) | 65 (35%) | 48 (41%) | 84% within last 12 months |
| Within last month | 248 (45%) | 79 (43%) | 47 (40%) | |
| Within the last 12 months | 121 (22%) | 34 (18%) | 20 (17%) | |
| Over 12 months ago | 12 (2%) | 6 (3%) | 3 (3%) | 15% |
| Usual primary care usage | Missing=88 | Missing=40 | Missing=17 | |
| At least once a month | 181 (33%) | 73 (40%) | 52 (45%) | - |
| At least once per 6 months | 285 (52%) | 79 (43%) | 45 (39%) | - |
| Once per 12 months or less | 84 (15%) | 31 (17%) | 18 (16%) | - |

*http://content.digital.nhs.uk/searchcatalogue?productid=24139&topics=1_2fWorkforce_2fSt aff+numbers&sort=Relevance&size=10&page=1#top
†https://www.ons.gov.uk/peoplepopulationandcommunity/populationandmigration/populationestimates/bulletins/ annualmidyearpopulationestimates/latest
GP, general practitioner; NHS, National Health Service; ONS, Office for National Statistics.

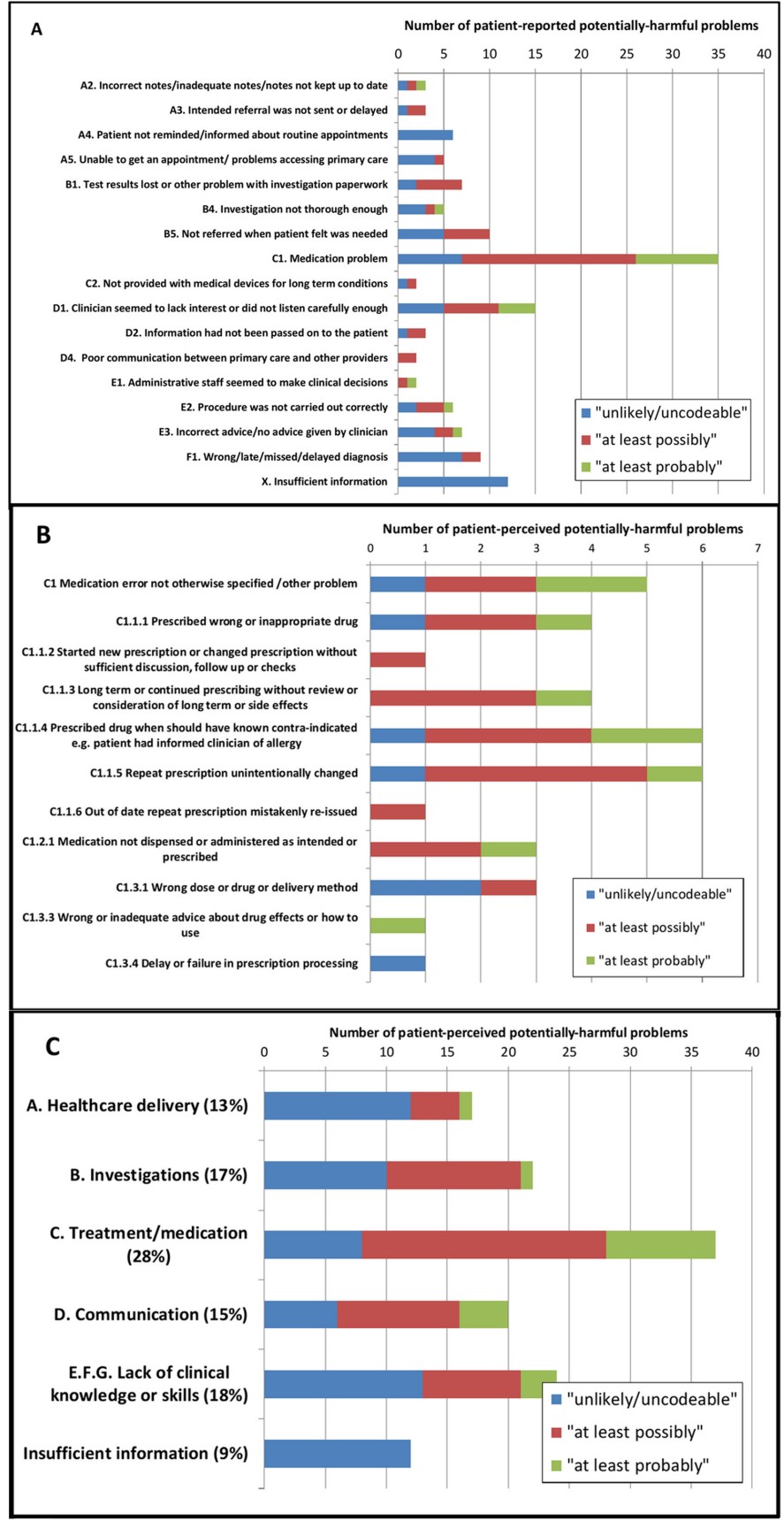

**Figure 1** Numbers of patient-perceived problems occurring in the last 12 months categorised by type of problem as described in Tables A & B, online Appendix 1 (A coded to 2 levels, B medication problems coded to 3 levels, C coded to 1 level). Colour coding describes clinican ranking as to the likelihood it is "probably" or "possibly" a potentially-harmful preventable problem as defined in Table 5.

respondents were recruited through the Help Beat Diabetes group (533, 84%, online supplementary appendix 1 box B). A flow chart of respondents through the survey is shown in online supplementary appendix 1 figure A; 223/638 (35%) of respondents reported ever experiencing a potentially harmful preventable problem in primary care of which 132 occurred within the past 12 months (21%). Sixty-two (10%) of these problems were not identified through the initial screening question (Q2) but required prompting through Q10 (box). A further 18 potentially harmful preventable problems involving friends or relatives where the respondent was present and occurred in the last 12 months were reported 13/418 (3%, online supplementary appendix 1 figure B).

### Characteristics of the respondents (aim 2)

The majority of respondents (592, 93%) had confidence and trust in the GP seen at their last appointment similar to the 2016 England proportion of 92% (Q1, box and table 1). Respondents were older than the UK generally, more likely to work or volunteer in the healthcare sector and tended to use primary care more frequently (table 1). Older respondents and those working or volunteering in the healthcare sector were no more likely to report a potentially harmful preventable problem occurring within the last 12 months but those using primary care more frequently were more likely to report a problem (table 2). There was a high response from healthcare professionals or volunteers (30% of respondents compared with approximately 3% of the UK adult population, table 1), but they were not more likely to report a preventable problem than non-healthcare workers/volunteers (35%, $P\chi^2=0.28$).

### The nature of the potentially harmful preventable problems (aim 2)

The types of patient-reported scenarios and their categorisation following clinician review are shown in figure 1. Medication-related problems were most frequently reported type of problem and also more likely to be ranked as a potentially harmful problem by clinicians, as were communication problems. The type of scenario categorised according to whether it arose from the open-ended screening question (Q2) or prompted through the list of potential problems (Q10) is shown in online supplementary appendix 1 figures C,D. Scenarios describing problems with appointments, accessing healthcare or loss of test

**Table 2** Prevalence of respondents reporting a potentially harmful preventable problem within the last 12 months and unadjusted and adjusted ORs estimated by logistic regression

| Respondent characteristics n=638 | Frequency— all reported n=132 | Unadjusted OR— all reports | Adjusted* OR— all reports | Adjusted* OR— after GP review (lower threshold, table 5) |
|---|---|---|---|---|
| Gender (87 missing) | | | | |
| Male | 53/283 (19%) | 1 (ref) | 1 (ref) | 1 (ref) |
| Female | 63/268 (24%) | 1.3 (0.9 to 2.0) | 1.4 (0.9 to 2.2) | 1.3 (0.7 to 2.3) |
| Age (85 missing) | | | | |
| 16–34 years | 11/42 (26%) | 1 (ref) | 1 (ref) | 1 (ref) |
| 35–54 years | 34/143 (24%) | 0.9 (0.4 to 1.9) | 0.8 (0.3 to 1.8) | 0.8 (0.3 to 2.1) |
| 55–64 years | 31/162 (19%) | 0.7 (0.3 to 1.5) | 0.7 (0.3 to 1.5) | 0.6 (0.2 to 1.7) |
| 65–74 years | 32/170 (19%) | 0.7 (0.3 to 1.4) | 0.6 (0.3 to 1.4) | 0.4 (0.2 to 1.2) |
| Over 75 years | 9/36 (25%) | 0.9 (0.3 to 2.6) | 1.1 (0.4 to 3.2) | 0.9 (0.2 to 3.2) |
| Last primary care contact (88 missing) | | | | |
| Within last week | 48/169 (28%) | 1 (ref) | 1 (ref) | 1 (ref) |
| Within last month | 47/248 (19%) | 0.6 (0.4 to 0.9) | 0.7 (0.4 to 1.1) | 0.6 (0.3 to 1.0) |
| Within the last 12 months | 20/121 (17%) | 0.5 (0.3 to 0.9) | 0.6 (0.3 to 1.2) | 0.5 (0.2 to 1.3) |
| Over 12 months ago | 3/12 (25%) | 0.8 (0.2 to 4.0) | 0.9 (0.2 to 4.2) | 0.4 (0.0 to 3.9) |
| Usual primary care usage (88 missing) | | | | |
| At least once a month | 52/181 (29%) | 1 (ref) | 1 (ref) | 1 (ref) |
| At least once per 6 months | 45/285 (16%) | 0.5 (0.3 to 0.7) | 0.6 (0.3 to 0.9) | 0.5 (0.3 to 0.9) |
| Once per 12 months or less | 18/84 (21%) | 0.7 (0.4 to 1.2) | 0.8 (0.4 to 1.6) | 0.7 (0.3 to 1.8) |
| Works or volunteers in healthcare (92 missing) | | | | |
| No | 72/380 (19%) | 1 (ref) | 1 (ref) | 1 (ref) |
| Yes | 41/166 (25%) | 1.4 (0.9 to 2.2) | 1.3 (0.8 to 2.1) | 1.5 (0.9 to 2.7) |

*Adjusted for gender, age, last primary care contact, usual primary care usage, works or volunteers in healthcare.
GP, general practitioner.

**Table 3** The patient's response to their perceived potentially harmful preventable problem and the primary care service involved for problems occurring in the last 12 months

| Primary care service | All reported problems | Clinician ranked 'possibly or higher' (lower threshold) |
|---|---|---|
| All services | 132 | 71 |
| GP surgery | 97 (73%) | 61 (86%) |
| Out of hours care/A&E/ambulance | 4 (3%) | 1 (1%) |
| Walk-in clinic | 2 (2%) | 0 |
| Dental surgery | 4 (3%) | 1 (1%) |
| Pharmacy | 7 (5%) | 6 (8%) |
| Community or district nursing | 4 (3%) | 0 |
| Opticians | 2 (2%) | 1 (1%) |
| Mental health services | 1 (1%) | 0 |
| Missing | 11 (8%) | 1 (1%) |
| **Did you discuss the problem with primary care staff?** | | |
| All respondents | 132 | 71 |
| Yes—discussed with primary care staff | 56 (42%) | 42 (59%) |
| No—did not discuss with primary care staff | 67 (51%) | 29 (41%) |
| Missing | 9 (7%) | 0 |
| **Reason not discussed with primary care staff** | | |
| All not discussing problem | 67 | 29 |
| Did not feel comfortable to discuss the problem | 16 (24%) | 8 (28%) |
| Could not find anybody with whom to discuss the problem | 21 (31%) | 10 (34%) |
| Unconcerned about the problem | 7 (10%) | 5 (17%) |
| Did not notice the problem at the time (or too ill) | 11 (16%) | 4 (14%) |
| Other | 5 (7%) | 2 (7%) |
| Missing | 7 (10%) | 0 |
| **Profession of discussant** | | |
| All discussing problem | 56 | 42 |
| GP | 28 (50%) | 19 (45%) |
| Practice manager | 5 (9%) | 5 (21%) |
| Receptionist | 2 (4%) | 1 (2%) |
| Practice nurse | 6 (11%) | 5 (12%) |
| Pharmacist or dispenser | 7 (13%) | 7 (17%) |
| General dental practitioner | 2 (4%) | 1 (2%) |
| Dietician | 1 (2%) | 1 (2%) |
| Missing | 5 (9%) | 3 (7%) |
| **Role of discussant in patient's care** | | |
| Member of staff directly involved | 23 (41%) | 16 (38%) |
| Another member of staff at same institution | 25 (45%) | 20 (48%) |
| Above unclear | 8 (14%) | 6 (14%) |

A&E, accident and emergency; GP, general practitioner.

results were more likely to arrive via the prompt question suggesting that patients did not see these as a potentially harmful problem in the first instance. The majority of potentially harmful preventable problems in the past 12 months occurred in general practice (73%, table 3) and pharmacy (5%, table 3).

### The patient's response to the potentially harmful preventable problem (aim 2)

Around half the respondents had not discussed their problem with anybody working in primary care (51%, table 3). The most common reasons for not discussing the problem were being unable to find a primary care

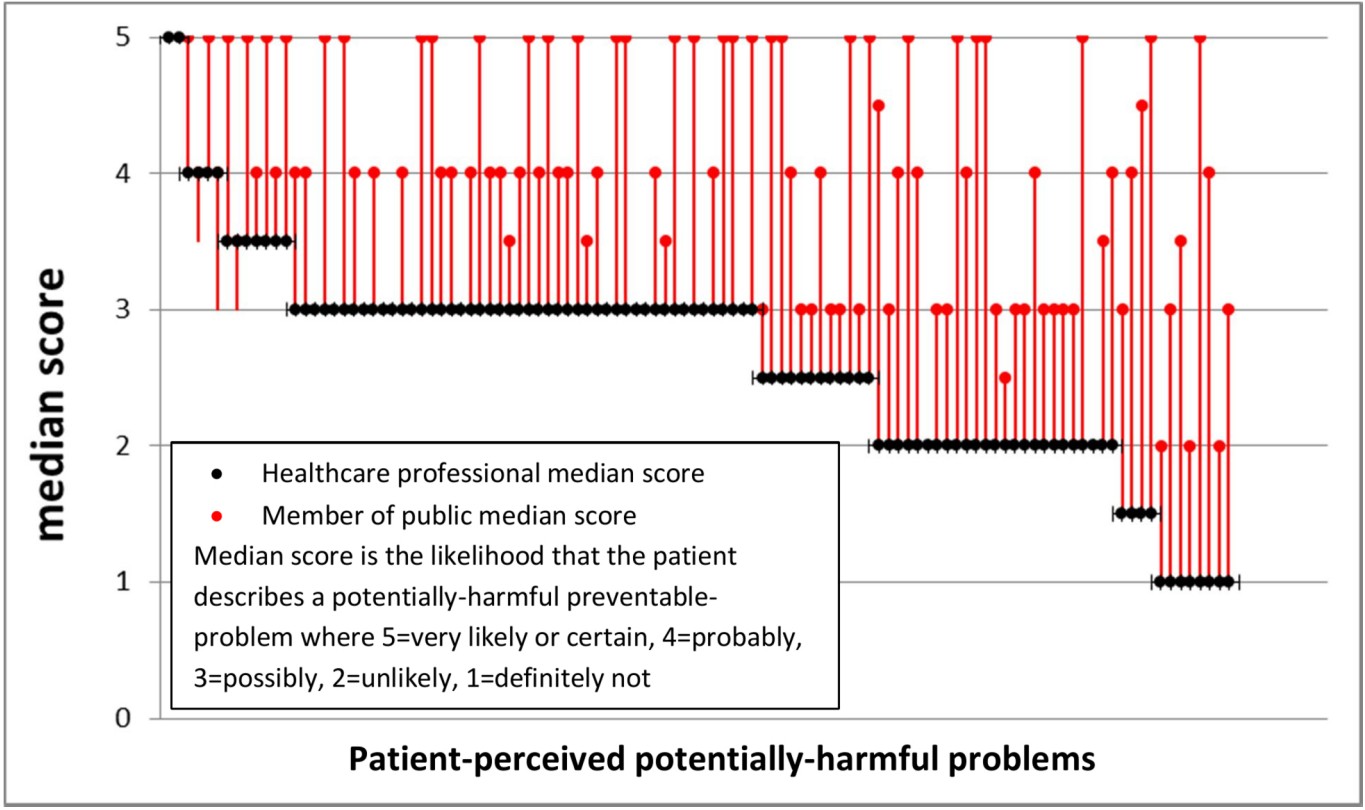

**Figure 2** Median estimates as to the likelihood that the patient describes a potentially harmful preventable problem occurring in the last 12 months by six clinicians and seven members of the public.

professional with whom to discuss the problem (31%, table 3) or they did not feel comfortable discussing their concerns (24%, table 3). The respondent's suggestions for ways to prevent the problem from happening are summarised in table 4. The most frequent suggestions were that clinicians should involve the patient more fully in the healthcare process (ie, listen to the patient and trust their judgement more) and be up to date with, and apply, the most recent information about the patient's condition (ie, take into account all of the patient's information—their medical history and results and letters).

### Likelihood the patient-reported scenario described a potentially harmful preventable problem (aim 3)

Generally, the members of the public assigned a higher probability to the likelihood that the patient-described scenario was a potentially harmful preventable problem compared with clinicians (figure 2, table 5). In 89/108 (82%) scenarios, the median score for the PPI researchers was higher than for the clinicians, and for 38 (35%) scenarios, the PPI median score was two or more points higher in a five-point scale. Following clinician review, 3% of the respondents were judged to have 'probably' experienced a potentially unsafe preventable problem during the past 12 months and 11% as 'possibly' (using higher and lower thresholds described in table 5). Scenarios described by healthcare professionals or volunteers were significantly more likely to be categorised as a

potentially harmful preventable problem following clinician review using both the lower (9% vs 16%, $P\chi^2=0.01$) and higher threshold (2% vs 6%, $P\chi^2=0.004$). Examples of the patient-reported scenarios with higher clinician rankings are shown in boxes 1–15, online supplementary appendix 2 and those with greatest disagreement between members of the public and clinicians in boxes 16–23, online supplementary appendix 2.

### DISCUSSION

We have designed and tested a survey to measure the frequency of occurrence of potentially harmful preventable problems in primary care and found it to be well understood and acceptable to patients. The open-ended questions (Q6–Q9, box) led to patient-described scenarios that mapped well to an existing taxonomy designed and used by clinicians and researchers (see online supplementary appendix 1[26 27]). This implies agreement between clinicians, researchers and patients in identifying the characteristics of a potentially harmful problem. Furthermore, the use of an open-ended screening question (Q2, box) to ensure that any problems unique to the patient perspective were identified did not find additional new types of problem. However, the open-ended question elicited more problems related to communication and medication suggesting that the public are more likely to view these as safety

**Table 4** Patient suggestions as to how the potentially harmful preventable problem might have been prevented

| How could it be prevented? | All reported problems n=132 | Clinician ranked 'possibly or higher' (lower threshold) n=71 |
|---|---|---|
| 1. More resources—all | 14 (11%) | 3 (4%) |
| 1.1 Quicker access to primary care | 7 (5%) | 2 (3%) |
| 1.2 More thorough and quicker investigations | 2 (2%) | 1 (1%) |
| 1.3 Fewer demands on primary care—more staff or fewer patients | 1 (1%) | 0 |
| 1.4 More time with clinicians for treatment and diagnosis | 2 (2%) | 0 |
| 1.9 Provision of resources to manage long-term conditions | 1 (1%) | 0 |
| 1.10 Provision of patient travel service for routine appointments | 1 (1%) | 0 |
| 2. Improved communication and involvement of patients | 26 (20%) | 18 (25%) |
| 2.1 Listen to the patient and trust their judgement more | 21 (16%) | 15 (21%) |
| 2.2 Tell patients about their diagnosis, test results, changes in medication or loss of results | 3 (2%) | 1 (1%) |
| 2.3 Improve communication between staff (within or outside primary care) | 2 (2%) | 2 (3%) |
| 3. Better organisation and administration | 17 (13%) | 10 (14%) |
| 3.1 Follow-up referrals and appointments to ensure they happen, be consistent in sending routine reminders | 10 (8%) | 3 (4%) |
| 3.2 Log in or process results as soon as received to avoid loss | 1 (1%) | 1 (1%) |
| 3.3 Keep the notes up to date, well-organised, safe and ensure information is transcribed accurately | 5 (4%) | 5 (7%) |
| 3.4 Keep a record of the location of equipment | 1 (1%) | 1 (1%) |
| 4. Improved prescribing systems | 18 (14%) | 17 (24%) |
| 4.1 More checks on prescribing and dispensing | 8 (6%) | 8 (11%) |
| 4.2 Check repeat prescriptions carefully, especially for transcribing errors | 8 (6%) | 7 (10%) |
| 4.3 Use medication reviews and computerised clinical decision support systems | 2 (2%) | 2 (3%) |
| 5. Better clinical practice | 19 (14%) | 10 (14%) |
| 5.1 Take in to account all the patient's information—their medical history and results and letters | 13 (10%) | 7 (10%) |
| 5.2 Address the patient's problem in some way—patients can feel their problem is being ignored | 5 (4%) | 2 (3%) |
| 5.3 Act on advice from other clinicians and test results | 1 (1%) | 1 (1%) |
| 6. Staff training | 11 (8%) | 7 (10%) |
| 6.1 More informed and better trained staff | 11 (8%) | 7 (10%) |
| Other responses | 27 (20%) | 6 (8%) |
| Do not know/missing | 21 (16%) | 3 (4%) |
| Problem was due to an individual member of staff | 2 (2%) | 1 (1%) |
| Prescribe right, better, different, more, less medicine | 1 (1%) | 0 |
| Better organisation | 1 (1%) | 0 |
| Laboratory procedures were the problem | 2 (2%) | 2 (3%) |

problems than problems related to appointments and referrals or investigations (see online supplementary appendix 1 figure C,D) in agreement with clinicians who were more likely to rank these types of scenarios as potentially harmful. The observation that members of the public were generally more likely to rank the scenarios as a potentially harmful preventable problem than clinicians (figure 2) is important;primary care should not only be safe but also be perceived as safe by patients.

### Strengths and weaknesses of the study
We believe that our survey captures the true patient perspective due to the involvement of members of the public as research partners through data acquisition to analysis and reporting in a codesigned study.

**Table 5** Categorisation of patient perceived potentially harmful preventable problems occurring in the last 12 months following review by clinicians and members of the public

| Group label | Threshold criteria | Clinician scores n=132 | Members of the public scores n=132 |
|---|---|---|---|
| 1. Higher threshold | Median score of 'very likely or certain' or 'probably' or at least one score of 'very likely or certain' | 18 (14%) | 87 (66%) |
| 2. Lower threshold | Median score of 'possibly' or at least one score of 'probably' or higher | 71 (54%) | 104 (79%) |
| 3. Any possibility | At least one score of 'unlikely' or higher | 106 (80%) | 109 (83%) |
| 4. No problem | All scores 'definitely not' or not coded | 1 (1%) | 0 |
| 5. Not coded | Insufficient information for coding by all coders | 25 (19%) | 23 (17%) |

By the use of a simple non-leading screening question, we encouraged respondents to express their own perspective on what constituted a potentially harmful preventable problem rather than directing them towards existing definitions. To ensure that we did not miss any problems, we followed up with a prompt that encouraged respondents to think in terms of the traditional view of patient safety problems. Furthermore, our survey goes further than describing and counting the frequency of occurrence of potentially harmful preventable problems and provides information about how patients dealt with the problem and how it could have been prevented that offers insight into ways to reduce the frequency of their occurrence. The absence of a link between practices and the patients allows for responses that might not occur if this survey were administered through the individual's practice. The main weakness of the study is the self-selection of the respondents who were older and tended to use primary care more frequently. More frequent users of primary care were more likely to report a problem, but age was not associated with the likelihood of reporting a problem. Our benchmarking question (Q1, box) showed that the respondents were similar to the English GP patient survey[25] in terms of their level of confidence and trust in their GP and not a group with a more negative attitude towards primary care as might have happened given the nature of the survey. We also acknowledge that, by design, this study is totally from the patient perspective. We aim to provide insight into the patient's perspective and not to imply that one or the other point of view is more important but rather there are differences in perceptions that need to be understood and reconciled.

**Strengths and weaknesses in relation to other studies**
Our finding that 35% of respondents perceived that they had experienced a potentially harmful problem in their lifetime is consistent with a European survey (43% of UK respondents felt that it was 'likely' that patients could be harmed by non-hospital healthcare).[17] This study offers some insight into the type of concerns that might underlie this apparent lack of confidence in primary care. A face-to-face interview in family practice waiting rooms in the USA reported that 16% of respondents believed a physician had made a mistake in their care.[28] The types of problem and patient responses to the problem are similar to those that have been described qualitatively,[1 22] but we have taken this a step further by quantifying their frequency of occurrence and other descriptors of the problem from the patient's perspective. In this small study, we did not find that patients were particularly likely to attribute blame to individual members of staff as has been observed previously,[3 4] perhaps partly due to the high proportion of respondents working or volunteering in healthcare.

**Unanswered questions and future research**
Our finding that 21% of respondents perceived that they had experienced a potentially harmful problem in the last 12 months, and the corresponding proportion following clinician review of 3% (higher threshold) to 11% (lower threshold) may well reflect the self-selected nature of the study population and needs to be validated in a large population level survey. We anticipate that a population-level survey would be fruitful since this approach yielded a number of patient-described scenarios that were amenable to further analysis including coding by clinicians. The high response to this pilot survey by healthcare professionals and volunteers probably reflects the population invited to complete the survey as well as an interest in this topic. It is likely that these respondents are better at articulating their potentially harmful problem given the higher ranking given by clinicians to scenarios originating from healthcare professionals. Healthcare professionals are an educated and accessible group with the expectations of a patient and with an understanding of the healthcare system who could provide a valuable resource for learning about preventable problems in primary care. Further work is also needed to understand and reconcile the differences between members of the public and clinicians' perceptions of a potentially harmful problem. In 1997, Professor Berwick stated 'The ultimate measure by which to judge the quality of a medical effort is whether it helps patients (and their families) as they see it. Anything done in healthcare that does not help a patient or

family is, by definition, waste, whether or not the professions and their associations traditionally hallow it.' If this tenet still holds, then we suggest there is a real need to influence patient's expectations and beliefs about primary care.

**Author affiliations**
[1]NIHR Greater Manchester Primary Care Patient Safety Translational Research Centre, Centre for Primary Care, Division of Population Health, Health Services Research and Primary Care, University of Manchester, Manchester, UK
[2]Research User Group (RUG) of the NIHR Greater Manchester Primary Care Patient Safety Translational Research Centre, Centre for Primary Care, Division of Population Health, Health Services Research and Primary Care, University of Manchester, Manchester, UK
[3]General Practitioner NHS Greater Glasgow and Clyde, Medical Directorate, NHS Education for Scotland, Glasgow, Scotland
[4]St Gabriels Medical Centre, Manchester, UK
[5]Central and South Manchester Specialty Training Programme for General Practice, Health Education England North West (HEENWE) Education and Research Centre, Wythenshawe Hospital, Manchester, UK
[6]Woodlands Dental Practice, Wirral, UK
[7]Medical Directorate, NHS Education for Scotland, Glasgow, UK
[8]Institute of Health and Wellbeing, University of Glasgow, Glasgow, UK

**Acknowledgements**  The authors would like to express their thanks and appreciation for the work done by Mary Aldred, Gitanjali Holt, Manoj Mistry, Carole Bennett and Lindsey Brown in coding the patient-described scenarios.

**Contributors**  SJS, AD, JB and CG conceived and designed the study. SJS, AD, JB, CG, AE, PB, JA, DT,SL, AD, RD and NM analysed the data. SJS wrote the manuscript, and is the guarantor. AD, JB, CG, AE, PB, JA, DT, SL, AD, RD, NM and SC edited the manuscript.

**Funding**  This study was funded by the National Institute for Health Research (http://www.nihr.ac.uk) through the Greater Manchester Primary Care Patient Safety Translational Research Centre (NIHR GM PSTRC), grant number gmpstrc-2012-1.

**Disclaimer**  The views expressed are those of the author(s)and not necessarily those of the NHS, the NIHR or the Department of Health. The funders had no role in study design, data collection and analysis, decision to publish or preparation of the manuscript.

**Competing interests**  All authors have completed the ICMJE uniform disclosure form at (available on request from the corresponding author).

**Patient consent**  Not required.

**Ethics approval**  University of Manchester Ethics Committee 2 (Approval 15372).

**Provenance and peer review**  Not commissioned; externally peer reviewed.

**Data sharing statement**  Raw data (coded only) is available from jill.stocks@manchester.ac.uk.

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
