## [Reviewer comments · BMJ Open]

ARTICLE DETAILS

TITLE (PROVISIONAL)	Development and piloting of a survey to estimate the frequency and nature of potentially-harmful preventable-problems in primary care from a UK patient's perspective
AUTHORS	Stocks, Susan; Donnelly, Ailsa; Esmail, Aneez; Beresford, Joanne; Gamble, Carolyn; Luty, Sarah; Deacon, Richard; Danczak, Avril; Mann, Nicola; Townsend, David; Ashley, James; Bowie, Paul; Campbell, Stephen

VERSION 1 – REVIEW

REVIEWER	Hilde Lurås Akershus University Hospital and University of Oslo
REVIEW RETURNED	28-Jun-2017

GENERAL COMMENTS	General comment The aim of the paper is to: 1) Design and pilot a survey to be used at the population level in UK primary care2) Explore the nature of the patient-perceived potentially-harmful preventable problems3) Present patient-suggested strategies for prevention4) Present clinicians' and the public's opinions regarding potential harm With that many objectives within one small paper it follows that the presentation within each objective is rather superficial and that the message to the reader (and the academic community as a whole) is difficult to grasp. In addition to 5 tables and one box the paper includes more than 20 pages with tables and scenarios in an appendix which are referred to throughout the paper, and this further makes it difficult to take in the message. From my point of view the paper is a mixture of a detailed report (because of all the necessary information included in the tables/appendix) and an unfinished (too scarce) paper. Design The design of the study seems ad hoc and casual. Why did for instance only three public members and one researcher found the group of "co-designers", and how was those four picked out. I am also curious regarding the 977 recruited from the organizations mentioned in Box A; why only these organizations and how many members could ideally have answered the questionnaire?
---

	It seems strange to report a response rate of 65 % when this is the percentage of the 977 that completed the survey after they have accessed the questionnaire, not the percentage of those that potentially could have responded/answered. Conclusion The paper needs more focus and a thorough review, and is not ready to be published yet.
--	--

REVIEWER	Charles Vincent University of Oxford UK
REVIEW RETURNED	08-Jul-2017

GENERAL COMMENTS	This paper reports the development and piloting of a survey to the frequency of potentially harmful and preventable events in primary care from the patient's perspective. The survey was co-designed with patients and piloted with 683 members of the general public, 21% of whom reported a potentially preventable event. The introduction to the paper effectively makes the case for examining patients' views of adverse events. We know that patients report different events from professionals to some degree and that their reports can be subsequently validated by clinicians. In primary care the perspective of the patient and family is particularly critical in bringing attention to bear on such issues as the coordination of care over time. The survey is commendably straightforward, though some aspects are a little problematic (discussed below). The approach taken of an initial open question followed by triggers of potential problems seems very good for eliciting the maximum information. General points The paper overall is well set out, clearly written and provides a generally comprehensive account of the study. The discussion is thoughtful in considering the strengths, limitations and implications of the study. There are however a number of places where the paper could be strengthened:  • The introduction is persuasive and makes important points. However, the second paragraph is somewhat muddled as all manner of different types of studies are briefly summarised and it is hard to know whether they are comparable. For instance, are all these studies in primary care settings? Are they all patient reported or are some estimates derived from other sources. The studies need to be more carefully explained and not just listed. • The survey asks about 'potentially harmful and preventable problems'. (Q2). It seems curious, and probably not optimal, to ask about two distinct and separate issues in one question. Surely it is necessary to ask separately about the occurrence of potentially harmful events (of whatever nature) and then separately ask about preventability (following the approach generally used in other studies of adverse events). Why was this approach taken? And will it be modified in future surveys?
--

	 • The Method states that five GPs, one dentist and one member of the public estimated the likelihood that the scenarios described were indeed 'potentially harmful and preventable problems'. How was this done? The problem noted above (confounding harm and preventability) must have made this quite difficult. What determined who reviewed each event? Did more than one of these people review each scenario? What criteria did they use? You simply state that they had to give a rating but give no information about what criteria were applied to each level of the scale. • The main point of the survey is to estimate the frequency of events and the overall percentage are given. However, it would be useful to provide more detail of the type of events reported which doesn't seem to appear in any of the tables. Given the fact that the paper argues that patients can provide a unique perspective it would be particular useful to say which of the events corresponded to the list of common events and which from the initial open question. Some examples of patient reported events would also be helpful, even as an appendix. I realise there may be limits on numbers of tables but the nature of the events reported, and the extent to which patients report novel events, seem particularly important. Minor points for clarification  • Abstract: the statement 'at least one clinician' estimated the likelihood is rather unclear. Does this mean other clinicians might have disagreed? • Abstract Trust and confidence in the GP? Does that mean that particular patient's trust or general trust in that GP? Presumably the former?
--	--

VERSION 1 – AUTHOR RESPONSE

Reviewer: 1

Hilde Lurås

Akershus University Hospital and University of Oslo Please state any competing interests or state 'None declared': None declared

General comment

The aim of the paper is to:

- 1) Design and pilot a survey to be used at the population level in UK primary care
- 2) Explore the nature of the patient-perceived potentially-harmful preventable problems
- 3) Present patient-suggested strategies for prevention
- 4) Present clinicians' and the public's opinions regarding potential harm

With that many objectives within one small paper it follows that the presentation within each objective is rather superficial and that the message to the reader (and the academic community as a whole) is difficult to grasp. In addition to 5 tables and one box the paper includes more than 20 pages with tables and scenarios in an appendix which are referred to throughout the paper, and this further makes it difficult to take in the message.

Author's response: Thank you for making this point. The focus of the paper is the development of a survey for use at the population level (point 1 above) to address points 2-4 above rather than actually addressing these aims. It is the pilot work to address these aims in a population level survey. We agree that we should have made this clearer in the specific aims section and for aims 2 and 3 we have prefaced the stated aim with "examine the potential of the survey to..." see P4 lines 4 & 8. We have re-organised the Appendices so that all the supplementary methods and results are in Appendix 1 in the order in which they are referenced in the paper and Appendix 2 only includes the example scenarios.

Reviewer 1: From my point of view the paper is a mixture of a detailed report (because of all the necessary information included in the tables/appendix) and an unfinished (too scarce) paper.

Author's response: We agree with the reviewer that we have necessarily covered a wide area in this paper. Our intention was to summarise our findings in the main paper but it was always going to be necessary to rely heavily on supplementary data to demonstrate the potential usefulness of the survey. We have made several changes in response to the reviewer's comments below as well as some changes to make the paper more readable.

Reviewer 1: Design

The design of the study seems ad hoc and casual. Why did for instance only three public members and one researcher found the group of "co-designers", and how was those four picked out.

Author's response: The study was always intended to be very much from the patient's perspective rather than an academic or clinician's perspective so in this work patients were very much equal partners. (The terms "patients" and "the public" tend to be used interchangeably because almost all of the population are eligible for NHS services.) The members of the public who are authors on this paper were members of the Greater Manchester Primary Care Patient Safety Translational Research Centre Research User Group (GMPSTRC RUG) and therefore had some experience of involvement in research. The study arose following informal discussions among the GMPSTRC RUG members and the researcher (SJS). The opportunity to be involved was offered to all 10 members of the RUG and 3 chose to become involved this project. If the study appears "ad hoc" it might be because it was driven largely by public input rather than a more traditional approach to research. Nonetheless we had a prospective study design and wrote a protocol/ethics application before beginning so it was not undertaken in a haphazard fashion (the protocol is available but would make the appendices even larger). An extra sentence (p3 lines 38-39) has been added to the introduction to explain more about the role of the GMPSTRC RUG in providing PPI.

Reviewer 1: I am also curious regarding the 977 recruited from the organizations mentioned in Box A; why only these organizations and how many members could ideally have answered the questionnaire?

It seems strange to report a response rate of 65 % when this is the percentage of the 977 that completed the survey after they have accessed the questionnaire, not the percentage of those that potentially could have responded/answered.

Author's response: Our aim was to trial the survey using as many members of the public as possible and we asked all the PPI networks for whom we had contact details to send out the questionnaire on our behalf. We were satisfied that 665 responses were sufficient to test our survey, if we had had a low number of responses we would have needed to find other ways to disseminate our survey. As some of the invited groups were organisations we had no way of knowing how they cascaded the invitation emails (if at all) therefore we had no way of knowing how many people had been invited to participate in the survey (except for one PPI group but there did not seem to be a reason to report a response rate in this group alone).

The only quantitative metric for response was the one we quoted i.e. the denominator was people who looked at the survey and therefore had the opportunity to read the patient information sheet. In the body of the text we had described this as the completion rate because, as the reviewer points out, it is not a response rate but we had neglected to correct this in the abstract where we had referred to a response rate. This has now been corrected in the abstract.

Reviewer 1: Conclusion

The paper needs more focus and a thorough review, and is not ready to be published yet.

Author's response: We have made several changes in response to the reviewers' comments and would be happy to make further changes in response to specific criticisms or suggestions from the reviewer.

Reviewer: 2

Charles Vincent

University of Oxford UK

Please state any competing interests or state 'None declared': None

Please leave your comments for the authors below

This paper reports the development and piloting of a survey to the frequency of potentially harmful and preventable events in primary care from the patient's perspective. The survey was co-designed with patients and piloted with 683 members of the general public, 21% of whom reported a potentially preventable event.

The introduction to the paper effectively makes the case for examining patients' views of adverse events. We know that patients report different events from professionals to some degree and that their reports can be subsequently validated by clinicians. In primary care the perspective of the patient and family is particularly critical in bringing attention to bear on such issues as the coordination of care over time. The survey is commendably straightforward, though some aspects are a little problematic (discussed below). The approach taken of an initial open question followed by triggers of potential problems seems very good for eliciting the maximum information.

Author's response: Thank you for the positive comments.

Reviewer 2: General points

The paper overall is well set out, clearly written and provides a generally comprehensive account of the study. The discussion is thoughtful in considering the strengths, limitations and implications of the study. There are however a number of places where the paper could be strengthened:

Author's response: Thank you for the positive comments.

Reviewer 2: • The introduction is persuasive and makes important points. However, the second paragraph is somewhat muddled as all manner of different types of studies are briefly summarised and it is hard to know whether they are comparable. For instance, are all these studies in primary care settings? Are they all patient reported or are some estimates derived from other sources. The studies need to be more carefully explained and not just listed.

Author's response: We agree that we could have done a better job in describing these papers. Few papers describe patient-reported incidence or frequencies of adverse events or safety problems. We cite 4 studies from the UK, USA, Spain and Norway and have added text to make it clear which studies are from the healthcare professional's perspective (the review papers references 10-12) or the patient's perspective (references 4, 17-19), see page 3, lines 16 to 32.

Reviewer 2: • The survey asks about 'potentially harmful and preventable problems'. (Q2). It seems curious, and probably not optimal, to ask about two distinct and separate issues in one question. Surely it is necessary to ask separately about the occurrence of potentially harmful events (of whatever nature) and then separately ask about preventability (following the approach generally used in other studies of adverse events). Why was this approach taken? And will it be modified in future surveys?

Author's response: Thank you for raising this point, it is important and we discussed it in detail at our project team meetings. We wished to design a screening question that would point the respondent towards the concept of "a mistake or a problem that could have been prevented" that had potential health implications i.e. we tried to describe the essence of a potentially harmful preventable problem so that the patient's own interpretation could come through from the start. We were concerned that if we asked about "a mistake or a problem" without further guidance they would answer with their actual health problem in mind (most people visit primary care because they have a health problem) then would respond it was not preventable at the next stage thereby not informing us about the type of problem we were interested in – the preventable problem. (Despite our efforts patients still occasionally responded with comments related to health problems e.g. prevention by losing weight or by taking better care of themselves and these were removed as not-preventable problems in this context.) Another factor was the number of questions required to identify the problem, we were concerned about the cost of the population-level survey and also potentially losing respondents through making it too long. Also we thought that a validated single screening question would have other uses besides estimating the incidence in a one-off survey such as surveillance of patient's perceptions over time. We did follow-up with the separate question about preventability (Q7) but it was intended to ensure that the respondent had reported a genuine preventable problem. In the majority of cases they backed up their first response by indicating that the problem was preventable suggesting that the first question was working as we had intended.

A population level survey was done shortly after the pilot survey and we did make some changes to the way the first question was asked but this was around the nature of the harm rather than the preventability. The version used in the population-level survey is shown below and will be published elsewhere.

Q2a. Have you experienced a situation with a primary care service where your health has ACTUALLY been made worse by a problem or error that could have been prevented?

1. Yes 2. No 3. Don't Know

Q2b. And have you experienced a situation with a primary care service where you SUSPECTED your health has been made worse by a problem or error that could have been prevented?

1. Yes 2. No 3. Don't Know

Q2c. And have you experienced a situation with a primary care service where your health could have been made worse had someone not NOTICED a problem or error?

1. Yes 2. No 3. Don't Know

Q2d. And have you experienced a situation with a primary care service where there was a problem or error that could have been prevented but it did not make your health worse?

1. Yes 2. No 3. Don't Know

Reviewer 2: • The Method states that five GPs, one dentist and one member of the public estimated the likelihood that the scenarios described were indeed 'potentially harmful and preventable problems'. How was this done? The problem noted above (confounding harm and preventability) must have made this quite difficult. What determined who reviewed each event? Did more than one of these people review each scenario? What criteria did they use? You simply state that they had to give a rating but give no information about what criteria were applied to each level of the scale.

Author's response: this information is provided in Table D in online Appendix 1 and pasted below. However we failed to signpost this from the paper and apologise for this omission. All the reviewers scored every scenario using the guidance in Table D below. They were asked "How likely do you think it is the patient was correct in thinking that their health might be worsened, or actually was made worse, because of a mistake or a problem in primary care that could have been prevented?" using the criteria described in the table below. They were also provided with a copy of the questionnaire and protocol. We accept that this method of scoring is not ideal but given the practical difficulties of obtaining specific information for clinical review we felt this was the best option of obtaining some level of prioritisation by clinicians as there was no alternative, at least at this stage of the survey development.

Table D. Scoring for likelihood that the patient-reported scenario is potentially-harmful preventable-problem

Score How likely do you think it is the patient was correct in thinking that their health might be worsened, or actually was made worse, because of a mistake or a problem in primary care that could have been prevented? Choose from the options below.

5 Very likely or certain (75-100% confident is a potentially unsafe scenario)

4 Probably (50-74% confident is a potentially unsafe scenario)

3 Possibly (25-49% confident is a potentially unsafe scenario)

2 Unlikely (bottom 25% confident is a potentially unsafe scenario)

1 Definitely not a potentially unsafe event (0% chance is a potentially unsafe scenario)

- Insufficient information

- Don't know

- Other - add text at end of row

Reviewer 2: • The main point of the survey is to estimate the frequency of events and the overall percentage are given. However, it would be useful to provide more detail of the type of events reported which doesn't seem to appear in any of the tables.

Author's response: This information is provided in Figure 2 and the coding system is described in Tables A&B, online Appendix 1. It is referred to in the results section on P7 lines 22-23. In the discussion section on p8 lines 7-10 it is pointed out that "The open-ended questions (Q6 to Q9, Box 1) led to patient-described scenarios that mapped well to an existing taxonomy designed and used by clinicians and researchers (Fig 2, Table A, online Appendix 1, 25, 26). This implies agreement between clinicians, researchers and patients in identifying the characteristics of a potentially-harmful problem."

Reviewer 2: Given the fact that the paper argues that patients can provide a unique perspective it would be particular useful to say which of the events corresponded to the list of common events and which from the initial open question.

Author's response: Two new figures have been added to Appendix 1 (Figures C&D) showing the distribution of the type of scenario according to route through the survey i.e. from the open question or the prompted question. This has been signposted in the results section p7 line 25 and described in the discussion p8 lines 8-10. The scenarios related to communication and medication were more likely to come via the open-ended question whereas scenarios related to the process of healthcare delivery such as delays in appointments and investigations were more likely to come via the prompt question. This suggests that patients are more concerned about problems related to communication and medication than appointments or referrals.

Reviewer 2: Some examples of patient reported events would also be helpful, even as an appendix. I realise there may be limits on numbers of tables but the nature of the events reported,

Author's response: All the scenarios ranked with a higher likelihood of being a potentially-harmful preventable-problem by clinicians and those with the greatest disagreement in ranking between clinician and members of the public are described in Appendix 2 alongside individual rankings. In total 23 example scenarios have been shown. This is signposted in the results section on p7 lines 18-19.

Reviewer 2: ...and the extent to which patients report novel events, seem particularly important.

Figure 2 shows that no novel events were reported. This was referred to in the discussion on p8 lines 6-7 "Furthermore, the use of a non-leading screening question (Q2, Box 1) to ensure that any problems unique to the patient perspective were identified did not find additional types of problem."

(However 12 events from the open-ended question could not be coded due to a lack of adequate information so it is possible these were novel events.)

Reviewer 2: Minor points for clarification

- Abstract: the statement 'at least one clinician' estimated the likelihood is rather unclear. Does this mean other clinicians might have disagreed?

Author's response: yes, this is a measure of the quality of the information collected in the survey. As mentioned above (Table D, Appendix 1) the clinician and members of the public had the option to indicate that the patient had not provided information of sufficient quality for them to form an opinion as to the likelihood the event described was a potentially-harmful preventable-problem. We are saying that for 82% of the reports at least 1 out of 6 clinicians deemed the information adequate to form an opinion. The actual number can be seen for some scenarios in boxes 1-23, Appendix 2.

To make this clearer P2, lines 18-20 now read ".....(82%) of these respondents provided a description that was adequate for at least one clinician to form an opinion about the potentially-harmful problem"

Reviewer 2: • Abstract Trust and confidence in the GP? Does that mean that particular patient's trust or general trust in that GP? Presumably the former?

Author's response: yes it means that particular patient's trust in the GP seen at their last appointment. It is the response to the question "Q1. Did you have confidence and trust in the GP you saw or spoke to at your last appointment?" which is also asked in the GP patient survey and was used as a bench marking question to check that the survey population was similar to that of England in terms of their confidence and trust in their GP. We felt that that this was important because a self-selected sample could be biased towards patients with negative opinions about primary care.

P2 line 21: We replaced "their GP" with "their own GP" to help make this clear

VERSION 2 – REVIEW

REVIEWER	Charles Vincent University of Oxford, UK
REVIEW RETURNED	23-Aug-2017

GENERAL COMMENTS	The authors have responded to the reviews with a comprehensive explanation of why they took certain decisions and pointing to a variety of materials in the appendices, some of which the reviewers probably missed on first reading due to the volume of material. I note that the Reviewer 1 considered that the paper needs more focus and a thorough review before publication. While the explanation and response to reviewers is commendably comprehensive the changes to the paper are modest and do not fully address the reviewers' concerns. I could enumerate several instances where I would still like more detail in the paper itself (such as giving more information about the alleged subject of the survey, preventable events). However it may be better to pose a different question and then invite the authors to consider whether they wish to make further changes. Like the first reviewer I found the paper tough going in that it took time to understand what was going on. Both reviews are I believe aimed at helping the authors improve the paper, rather than simply asking for a justification of what was done. At the moment the paper is still quite difficult to follow and much critical information remains in the appendices. Reviewers are more patient than readers and more willing to try to understand authors' intentions; readers will simply abandon a paper which doesn't appear to provide critical information. Do the authors simply want to post this information and tables online and make them available or do they want people to read and understand the paper? If the latter, then it might be worth looking again at the reviews and considering how to make the paper flow more smoothly, how to incorporate the information the reviewers felt was lacking (rather than continually directing us to appendices) and bringing some of the thinking in the response to reviews into the actual paper. In summary, I would say the authors have provided a satisfactory technical response in terms of their justification to the reviewers. However, there is no doubt that with a more substantial revision the paper could be made a great deal more readable and understandable.
--

VERSION 2 – AUTHOR RESPONSE

Authors' response: Thank you to the reviewer for commenting on this paper again. We appreciate the helpful comments and acknowledge that we should have made more substantial changes in the first revision. We thank the reviewers for their patience.

Reviewer's comment: The authors have responded to the reviews with a comprehensive explanation of why they took certain decisions and pointing to a variety of materials in the appendices, some of which the reviewers probably missed on first reading due to the volume of material. I note that the Reviewer 1 considered that the paper needs more focus and a thorough review before publication. While the explanation and response to reviewers is commendably comprehensive the changes to the paper are modest and do not fully address the reviewers' concerns.

Authors' response: We agree that we could have done a better job of guiding the reader through our study in our original submission. We re-organised our appendices substantially as well as improving the sign posting to make the paper easier to read in the re-submission which would not be visible as track changes but nonetheless we did make substantive changes. We have now further revised the manuscript very carefully and made substantial changes that can be seen in the marked copy. We have also labelled each section of the methods and results with the corresponding aim. We hope that it is now much easier to read.

Reviewer's comment: I could enumerate several instances where I would still like more detail in the paper itself (such as giving more information about the alleged subject of the survey, preventable events). However it may be better to pose a different question and then invite the authors to consider whether they wish to make further changes.

Authors' response: We are sorry that we do not fully understand this comment. Perhaps the reviewer is referring back to their original comment below?

Reviewer's comment from 1st review: "The survey asks about 'potentially harmful and preventable problems'. (Q2). It seems curious, and probably not optimal, to ask about two distinct and separate issues in one question. Surely it is necessary to ask separately about the occurrence of potentially harmful events (of whatever nature) and then separately ask about preventability (following the approach generally used in other studies of adverse events). Why was this approach taken? "

Further comments from the authors: Perhaps we should add to our previous response given that, as the reviewer points out, we took a rather different approach to previous work on preventable events. This is a direct consequence of the co-design nature of the study and the question was framed in this way because of the input of the members of the public. Before designing the question the PPI group discussed the questions used in other surveys with similar aims and felt that the questions asked were just not framed in a way that would elicit recognition of the type of event or problem that we were looking for or the information that the public would wish to convey. Also they felt other surveys were too long and too focussed on tick boxes that could mislead. Hence they favoured giving the respondent the freedom to make their own interpretation of what a preventable problem actually was and then deal with the consequences of the imperfect information in the analysis. A trained researcher would always approach the question design from the point of view of facilitating the analysis that would follow, so yes it would be sensible to separate the 2 concepts, problem and preventable but, as we have explained, this did not provide us with the response we needed. The point of involvement was to listen to the members of the public in advising the best way to get the information we needed.

Essentially the members of the public felt that they had not been asked the right question in previous professionally-designed surveys. We have added a new paragraph about this as the first paragraph in the results p6 lines 11-23.

If this is not what the reviewer was intending us to discuss we are happy to revisit this comment.

Reviewer's comment: Like the first reviewer I found the paper tough going in that it took time to understand what was going on. Both reviews are I believe aimed at helping the authors improve the paper, rather than simply asking for a justification of what was done.

Authors' response: We appreciate the reviewer's comments and have tried our best to make the paper more readable. We apologise for not responding to this in our first revision and thank the reviewer's for their patience.

Reviewer's comment: At the moment the paper is still quite difficult to follow and much critical information remains in the appendices.

Authors' response: We would really like to put more information into the main paper but are limited by the journal's requirements. As stated above we have thoroughly reviewed the paper to try and make it more readable by discussing and describing the contents of the appendices more fully.

Reviewer's comment: Reviewers are more patient than readers and more willing to try to try to understand authors' intentions; readers will simply abandon a paper which doesn't appear to provide critical information. Do the authors simply want to post this information and tables online and make them available or do they want people to read and understand the paper? If the latter, then it might be worth looking again at the reviews and considering how to make the paper flow more smoothly, how to incorporate the information the reviewers felt was lacking (rather than continually directing us to appendices) and bringing some of the thinking in the response to reviews into the actual paper.

Authors' response: As we explain this is the pilot work for a population level survey. We do need to publish this information separately from the paper describing the population level survey because of the space limitations of most journals. However we also have some findings that are unique to this survey and agree with the reviewer that we really should have made a better job of describing this work. We have undertaken a thorough review and hope it is now a more coherent piece of work.

Reviewer's comment: In summary, I would say the authors have provided a satisfactory technical response in terms of their justification to the reviewers. However, there is no doubt that with a more substantial revision the paper could be made a great deal more readable and understandable.

Authors' response: Once again thank you for pointing this out and we have taken this criticism fully on board and revised the paper accordingly.

VERSION 3 – REVIEW

REVIEWER	Charles Vincent Department of Psychology University of Oxford
REVIEW RETURNED	25-Oct-2017
GENERAL COMMENTS	The authors have patiently and comprehensively responded to the further requests for improvement and clarification of the manuscript. The restructured and expanded results section in particular is hugely improved and now conveys the findings clearly to the reader. The paper can now be read and understood in its own right which should encourage readers to seek more depth and detail in the appendices.